# Evaluating ChatGPT’s Concordance with Clinical Guidelines of Ménière’s Disease in Chinese

**DOI:** 10.3390/diagnostics15162006

**Published:** 2025-08-11

**Authors:** Mien-Jen Lin, Li-Chun Hsieh, Chin-Kuo Chen

**Affiliations:** 1Department of Medical Education, Chang Gung Memorial Hospital, Taoyuan 33305, Taiwan; j565243782@gmail.com; 2Department of Otolaryngology-Head and Neck Surgery, Mackay Memorial Hospital, Taipei City 10449, Taiwan; lichunhsieh1978@gmail.com; 3Department of Audiology and Speech Language Pathology, Mackay Medical University, New Taipei 25245, Taiwan; 4Department of Medicine, Mackay Medical University, New Taipei 25245, Taiwan; 5Department of Otolaryngology-Head and Neck Surgery, Chang Gung Memorial Hospital, Keelung 20401, Taiwan; 6School of Traditional Chinese Medicine, College of Medicine, Chang Gung University, Taoyuan 33302, Taiwan; 7Department of Otolaryngology-Head and Neck Surgery, Chang Gung Memorial Hospital, Taoyuan 33305, Taiwan

**Keywords:** generative AI, ChatGPT, Ménière’s disease, clinical practice guidelines, healthcare, Chinese

## Abstract

**Background**: Generative AI (GenAI) models like ChatGPT have gained significant attention in recent years for their potential applications in healthcare. This study evaluates the concordance of responses generated by ChatGPT (versions 3.5 and 4.0) with the key action statements from the American Academy of Otolaryngology–Head and Neck Surgery (AAO-HNS) clinical practice guidelines (CPGs) for Ménière’s disease translated into Chinese. **Methods**: Seventeen questions derived from the KAS were translated into Chinese and posed to ChatGPT versions 3.5 and 4.0. Responses were categorized as correct, partially correct, incorrect, or non-answers. Concordance with the guidelines was evaluated, and Fisher’s exact test assessed statistical differences, with significance set at *p* < 0.05. Comparative analysis between ChatGPT 3.5 and 4.0 was performed. **Results**: ChatGPT 3.5 demonstrated an 82.4% correctness rate (14 correct, 2 partially correct, 1 non-answer), while ChatGPT 4.0 achieved 94.1% (16 correct, 1 partially correct). Overall, 97.1% of responses were correct or partially correct. ChatGPT 4.0 offered enhanced citation accuracy and text clarity but occasionally included redundant details. No significant difference in correctness rates was observed between the models (*p* = 0.6012). **Conclusions**: Both ChatGPT models showed high concordance with the AAO-HNS CPG for MD, with ChatGPT 4.0 exhibiting superior text clarity and citation accuracy. These findings highlight ChatGPT’s potential as a reliable assistant for better healthcare communication and clinical operations. Future research should validate these results across broader medical topics and languages to ensure robust integration of GenAI in healthcare.

## 1. Introduction

Ménière’s disease (MD) is a chronic inner ear disorder characterized by episodic vertigo, fluctuating sensorineural hearing loss, tinnitus, and aural fullness [1]. It predominantly affects individuals aged 40 to 60, with a prevalence of 0.3 to 1.9 per 1000 people and a slight predominance in females [2,3]. Although the etiology remains uncertain, a combination of genetic and environmental factors, alongside endolymphatic hydrops—an excessive accumulation of fluid in the inner ear—plays a critical role in its pathophysiology [4,5]. Current treatments emphasize symptom management through dietary modifications, medications, and surgical interventions in severe cases [5,6]. However, MD often progresses, resulting in permanent auditory and balance impairments that significantly reduce quality of life [7,8,9].

Generative artificial intelligence (GenAI) has made remarkable advancements, driven by improvements in machine learning, computational power, and data storage [10]. GenAI models can generate high-quality text, images, and other content, offering innovative solutions in medicine, such as enhanced brain MRI segmentation, drug design, and diagnostic support [11,12,13,14,15]. ChatGPT, one such GenAI model, has gained popularity for its ability to translate complex medical information into accessible language, aiding both clinicians and patients in understanding intricate healthcare topics [16,17]. Despite its potential, ChatGPT’s accuracy in specialized fields like otology and its performance in languages other than English remain concerns [18]. This study evaluates the concordance of ChatGPT (versions 3.5 and 4.0) responses with the American Academy of Otolaryngology–Head and Neck Surgery (AAO-HNS) clinical practice guidelines (CPGs) for MD, translated into Chinese. The primary objective is to assess ChatGPT’s reliability as a source of medical information and compare the textual and performance differences between the two versions. By addressing these questions, this study aims to provide insights into whether ChatGPT is a reliable tool for enhancing healthcare communication and clinical workflows.

## 2. Materials and Methods

This study utilized ChatGPT’s question-and-answer interface to evaluate its concordance with clinical practice guidelines for MD. A total of 17 questions were designed based on the key action statements (KASs) from the AAO-HNS guidelines. These KASs provide specific recommendations or discouragements for clinical practices, forming the basis for the questions.

### 2.1. Question Design and Prompting

To ensure clarity and relevance, most questions were structured as yes/no queries, with some including one to two sub-questions for enhanced coherence. For instance, a question derived from KAS 3 asked, “Should clinicians carry out audiometric tests on all patients with a suspected clinical diagnosis of MD?” This direct approach minimized ambiguity and enabled precise evaluation of ChatGPT’s responses. The designed questions were then posed to ChatGPT 3.5 and 4.0 with both versions generating 17 responses, for further comparative analysis.

The responses of the models in the Chinese linguistic context were jointly verified by three authors to ensure their accuracy. Subsequently, the performance of ChatGPT-3.5 and ChatGPT-4.0 was compared, with a focus on analyzing potential discrepancies between the models’ outputs in Chinese and the intended meaning in English. This evaluation approach seeks to determine whether the performance variations observed between ChatGPT-3.5 and ChatGPT-4.0 in the Chinese context are attributable to inherent linguistic characteristics.

### 2.2. Response Review and Grading

All responses were independently reviewed by the authors, with consensus reached regarding alignment with the AAO-HNS guidelines. Inter-rater reliability prior to reaching consensus was calculated using Cohen’s Kappa with Microsoft Excel. Responses were classified into four categories:Correct: fully adhered to the guidelines, including timeframe, recommended actions, and treatment options.Partially Correct: aligned with guidelines but omitted key details.Incorrect: contradicted the guidelines.Non-answer: restated the question or provided irrelevant information.

The full list of questions and ChatGPT responses can be found in Appendix A (original version) and Appendix A (translated version).

### 2.3. Data Analysis

Correctness rates were calculated for all questions, including sub-questions. Two sets of responses were generated for comparison:ChatGPT-3.5 model (Chinese).ChatGPT-4.0 model (Chinese).

Comparative analysis was conducted to evaluate performance differences between the two versions. To ensure unbiased results, a new chatbox was opened for each query, minimizing the influence of ChatGPT’s contextual memory from previous interactions.

### 2.4. Statistical Evaluation

Data organization was conducted in Microsoft Excel, and statistical analyses were performed using IBM SPSS Statistics version 27. Fisher’s exact test was applied to compare correctness rates between models, with statistical significance defined as *p* < 0.05 for two-sided tests.

### 2.5. Ethical Considerations

As no patient medical records were used, all data were generated through hypothetical scenarios based on the guidelines. This study was approved by the Chang Gung Medical Foundation Institutional Review Board (No. 202500464B1). A descriptive flowchart of the study design is provided in Figure 1.

## 3. Results

### Concordance with the Guideline

ChatGPT-generated responses were evaluated across 17 questions asked in Chinese using two models, ChatGPT-3.5 and ChatGPT-4o. ChatGPT-3.5 produced 14 correct answers (82.4%), 2 partially correct answers (11.8%) and 1 non-answer (5.9%), while ChatGPT-4o generated 16 correct answers (94.1%) and 1 partially correct answer (5.9%). Neither model provided incorrect responses. Combined, the two models generated 30 correct answers out of 34 responses (88.2% correctness rate), 3 partially correct answers (8.8%), and 0 incorrect answers, resulting in a 97.1% correctness rate when combining correct and partially correct responses. ChatGPT-4.0 outperformed ChatGPT-3.5 with a higher proportion of correct answers (94.1% vs. 82.4%). However, the difference in correctness rates was not statistically significant based on Fisher’s exact test (*p* = 0.6012), indicating that both models demonstrated high accuracy and that ChatGPT-3.5’s performance was non-inferior. Inter-rater reliability prior to reaching consensus was calculated to be 0.65. A descriptive comparison of correctness rates is illustrated in Figure 2.

The ChatGPT-4.0 model demonstrated a clear advantage in providing references, explicitly citing authoritative sources such as the AAO-HNS guidelines and the international classification for vestibular and headache disorders (ICHD) in its responses. These references were specifically mentioned when addressing questions about diagnostic criteria, underscoring its ability to align its answers with recognized guidelines and enhancing the credibility of its outputs. In contrast, the ChatGPT-3.5 model did not cite any references in its replies, which may limit its perceived authority and adherence to evidence-based practices. Both models displayed empathy in addressing questions regarding clinician–patient communication. For instance, when asked, “Should clinicians educate patients with Ménière’s disease (MD) about the natural history, measures for symptom control, treatment options, and outcomes?”, ChatGPT-3.5 recommended promoting open and honest communication, emphasizing personalized information to help patients make informed decisions tailored to their specific circumstances. ChatGPT-4.0, however, went a step further by stressing the importance of presenting information in a clear and accessible manner, encouraging patients to ask questions, and ensuring they feel reassured and in control of their care. Moreover, when responding to a question adapted from the seventh KAS in the guidelines, ChatGPT-3.5 required a follow-up query to provide a complete and accurate answer, whereas ChatGPT-4.0 addressed the question correctly on the first attempt, as illustrated in Appendix A. These distinctions highlight ChatGPT-4.0’s superior ability to integrate evidence-based references, deliver empathetic and patient-centered responses, and accurately respond to guideline-based inquiries.

Many of the responses emphasized the importance of seeking professional help, such as consulting otolaryngologists or physical therapists. ChatGPT-4.0 showed a greater ability to reference specific medical specialties, with eight mentions compared to ChatGPT-3.5’s two mentions (Figure 3). This suggests that the 4.0 model is more effective in directing patients to the appropriate specialists for their concerns, potentially streamlining the diagnostic and treatment process. Additionally, ChatGPT-3.5 tended to provide mixed and less structured paragraphs, whereas ChatGPT-4.0 excelled in enumerating and organizing relevant details, resulting in clearer and more structured responses. However, a notable limitation of ChatGPT-4.0 was its tendency to include excessive details unrelated to the questions, which could overwhelm patients with redundant information and reduce the overall readability of its responses. A thorough comparison of ChatGPT-3.5 and 4.0 is demonstrated in Table 1.

## 4. Discussion

MD is a chronic inner-ear disorder characterized by hearing and balance problems, predominantly affecting middle-aged individuals. In Chinese-speaking societies such as Taiwan and China, specific prevalence data on MD remain scarce. However, studies from Japan estimate the prevalence to range from 17 to 34.5 cases per 100,000 individuals, significantly lower than prevalence rates reported in European populations, which can reach up to 200 cases per 100,000 individuals. This disparity may be influenced by genetic predispositions, environmental factors, and differences in diagnostic criteria. MD is strongly associated with vertigo-related anxiety and psychiatric comorbidities, including anxiety disorders, phobias, and depression. Notably, psychiatric symptoms tend to be more pronounced during periods of vestibular excitation and are less troublesome when peripheral vestibular function is diminished [19,20]. Chronic vestibular dysfunction negatively impacts patients’ quality of life and has been linked to potential cognitive decline [21]. The persistent discomfort arising from the unpredictable and chronic nature of MD, coupled with the absence of a definitive cure, often leads patients to seek additional information and support online.

This study is the first to incorporate Chinese into a comparative analysis of ChatGPT’s performance between the 3.5 and 4.0 models on the topic of MD. To the best of our knowledge, no comprehensive, evidence-based clinical practice guidelines for MD are currently available in Chinese [22]. To address this gap, we translated the KASs from the MD chapter of the clinical practice guidelines published by the AAO-HNS into Chinese and developed corresponding prompts based on these translations. The AAO-HNS CPG for MD is a highly regarded resource, providing evidence-based recommendations authored by leading experts in otolaryngology. Its primary objective is to improve diagnostic accuracy while reducing unnecessary diagnostic tests and imaging [5]. These guidelines synthesize the best available clinical evidence into clear and structured recommendations, offering clinicians practical tools to enhance decision-making processes. This study evaluates ChatGPT-generated responses to assess their alignment with these guidelines, their accuracy, and their ability to assist with healthcare delivery, as well as their potential utility as supplementary tools in medical education and patient counseling. In addition, we also attempted to pose the predetermined questions on DeepSeek R1, a rapidly growing GenAI platform known for its capabilities comparable to OpenAI’s GPT models. DeepSeek’s emergence, driven by an advanced language model built with cost-effective chips, showcases the swift evolution of GenAI while also raising concerns about potential personal data leaks and the transparency of its training data sources. When queried with MD-related questions, DeepSeek can provide thorough responses with clarity and reputable references, such as recommendations from the AAO-HNS CPG or the Headache Classification Committee of the International Headache Society. However, frequent server overload alerts prevented us from completing the study and ultimately forced us to discontinue further investigation into DeepSeek’s performance (Appendix A). If issues related to server overload and user privacy are resolved, further studies can be conducted on DeepSeek to validate the reproducibility of previous research.

Overall, our study revealed no significant difference in correctness rates between the ChatGPT-4.0 and 3.5 models, indicating comparable performance in terms of accuracy. Given its high correctness rate, ChatGPT demonstrates strong potential as an online medical counsellor, capable of providing real-time, accurate, and professional responses. However, notable distinctions were observed between the two models. The ChatGPT-4.0 model consistently delivered more comprehensive and detailed replies, often including references and enumerating key points directly relevant to the questions. In contrast, the 3.5 model occasionally produced vague or off-topic answers, which could lead to confusion or misinformation for both patients and healthcare workers. Additionally, the 4.0 model more frequently emphasized the importance of consulting healthcare professionals, particularly otolaryngologists (ENT doctors), thereby guiding users toward appropriate medical care. Another key advantage of the 4.0 model was its superior fluency in Chinese, making it more accessible and effective for Chinese-speaking users. These nuanced differences highlight the 4.0 model’s enhanced utility in delivering user-centric and contextually appropriate medical guidance.

Chinese and English are significantly different in their written forms, including characters, word formation, syntactic and morphological complexity, and overall sentence and paragraph structure. In Chinese, characters serve as the fundamental units of writing, typically representing a morpheme—the smallest unit of meaning. For English texts, established readability indices like the Flesch Reading Ease Score (FRES) and Flesch–Kincaid Grade Level (FKGL) are standard tools for assessing text difficulty. In prior Chinese-language health literacy studies, the Chinese readability index explorer (CRIE) has been used to provide readability scores for certain texts, such as website information regarding breast cancer [23]. However, there is currently no standardized Chinese readability score, and the online CRIE readability score was also not available throughout our study. Therefore, we decided not to include the Chinese readability assessment in our study.

Numerous studies have compared ChatGPT’s responses to established clinical guidelines; however, none have specifically focused on MD in Chinese [17,24,25,26,27,28,29,30,31]. Most of these studies utilized internal or external reviewers to assess the accuracy and validity of the generated content. One study found that ChatGPT’s responses to allergic rhinitis-related topics were, unexpectedly, more accurate in Chinese than in English for both GPT-3.5 and GPT-4. However, this pattern was only observed in GPT-4 when addressing chronic rhinitis-related queries [31]. Another study demonstrated that ChatGPT-4 achieved an overall accuracy rate of 75% across all subjects in the Taiwan Audiologist Qualification Examination, despite the questions being presented in a mix of Chinese and English [32]. Additionally, research has evaluated ChatGPT’s performance on sudden sensorineural hearing loss-related questions in Korean, using both the textbook from the Korean Society of Otorhinolaryngology-Head and Neck Surgery and the AAO-HNS clinical guidelines as benchmarks [29]. These findings underscore the importance of further studies to evaluate the performance of AI-generated responses in languages beyond English and Chinese, as expanding linguistic validation is crucial for global applicability and inclusivity.

Despite these strengths, significant limitations have also been identified. One study emphasized that ChatGPT has notable gaps in accuracy compared to reputable reference sites, making it unsuitable as a primary source for patients and clinicians seeking medical information [30]. Another study reported that while ChatGPT-4.0 outperformed ChatGPT-3.5 in terms of comprehensiveness and accuracy when addressing MD-related queries, neither version met readability standards for average patients [28]. Additionally, a comparative study observed that both ChatGPT and Bard (later rebranded as Gemini, a GenAI chatbot by Google) frequently recommended consulting medical professionals but lacked specificity and consistency in triage categorization. This inconsistency in prioritizing treatment based on condition severity often led to patient confusion [27]. These findings underline the potential of AI tools in healthcare while also highlighting the importance of addressing their current limitations.

The consensus across these studies is that AI tools like ChatGPT demonstrate significant potential in various domains, including advancing medical education, improving patient comprehension, and supporting initial diagnostic processes in otolaryngology. The rapid evolution of AI models has ushered humanity into a new era, where the processing of vast amounts of medical information and data is accomplished with unprecedented speed and efficiency. ChatGPT, with its question-and-answer format, serves as a powerful aid in healthcare, seamlessly integrating into existing clinical workflows to provide real-time feedback [33]. Its ability to summarize specialized medical knowledge efficiently and offer additional context when needed makes it a valuable tool for healthcare professionals. Furthermore, ChatGPT can generate initial differential diagnoses, customize responses for specific clinical scenarios, and quickly access relevant information, positioning it as a potentially indispensable assistant for healthcare workers in diagnostic processes [34].

However, concerns about the development and application of GenAI remain significant. Notable performance drifts have been observed in GPT-3.5 and GPT-4 over relatively short periods, highlighting the need for continuous monitoring and evaluation to ensure their reliability and effectiveness in real-world applications. Furthermore, how ChatGPT models are updated lacks transparency, leaving users without a clear understanding of the causes behind these behavioral drifts [35]. To fully harness GenAI’s potential while safeguarding patient safety and optimizing healthcare delivery, persistent refinement of AI capabilities must be accompanied by professional oversight.

The necessity for professional oversight is a recurring theme in the literature. Despite ChatGPT’s high accuracy in certain areas, it cannot replace professional medical evaluations due to its occasional “hallucinated” responses, which have the potential to mislead patients [25,26]. This concern is echoed by studies concluding that, while AI tools like ChatGPT show promise in providing medical information to patients and healthcare providers, they are not yet dependable enough for critical tasks such as test-taking or clinical decision-making without professional guidance. Issues such as variability in responses and the lack of integration with reputable reference sources further emphasize the importance of professional supervision in GenAI’s application [27,30]. Misinformation provided by GenAI models may cause extra time and costs for both accessing and delivering appropriate healthcare service. Delay of treatment can lead to undesired consequences, which can be fatal or irreversible in some cases. In our study, the “non-answer” response generated by the 3.5 model may cause confusion for the users, resulting in wasted time and effort in searching for answers. The “partially correct” responses may have less negative impact for patients, but healthcare workers could miss crucial details regarding the queried medical information.

This study has several limitations. First, the analysis was conducted solely on the ChatGPT platform, excluding comparisons with other generative AI tools, such as Gemini by Google. Future studies can apply a similar research approach to various other or even better Gen-AI platforms and models. Second, the questions were specifically focused on MD, which may limit the generalizability of the findings to other medical domains. Further research in diverse fields is needed to extensively assess ChatGPT’s capabilities. Furthermore, the primary reference used in this study was the AAO-HNS clinical practice guidelines, which may not be universally applicable across different countries and healthcare systems. However, the AAO-HNS CPG provide the most comprehensive information with evidence-based statements on the topic of MD among the currently available resources. Finally, increasing reading ease of the medical information is fundamental in facilitating patient education. Although some previous studies assess the readability of both guideline content and GenAI-generated texts, this study does not conduct a readability assessment due to the unavailability of credible indices or scores for evaluating readability in Chinese. Nonetheless, all the queries were designed as yes/no questions, which may lower the difficulty thresholds for both reading and understanding for patients to a certain extent.

## 5. Conclusions and Future Direction

This study highlights the capability of the ChatGPT model to deliver accurate medical information consistent with the KASs in the AAO-HNS CPG on MD. No significant difference in correctness rates was observed between ChatGPT-3.5 and 4.0 in Chinese. However, contextual differences were noted, with ChatGPT-4.0 being more likely to provide detailed, structured responses that enumerate major points and emphasize the importance of consulting healthcare professionals, such as otolaryngologists. Although prior studies have emphasized the necessity of professional oversight, our findings suggest that the evolving GenAI models hold great potential as health professional consultants, particularly in enhancing the efficiency and quality of healthcare delivery. Future research should aim to validate the reproducibility of these findings by applying the methodology to other medical domains and additional languages. Such efforts will be essential in improving the reliability, consistency, and practical applicability of GenAI, advancing its role in delivering appropriate medical guidance while maintaining high standards of safety and accuracy.

## Figures and Tables

**Figure 1 diagnostics-15-02006-f001:**
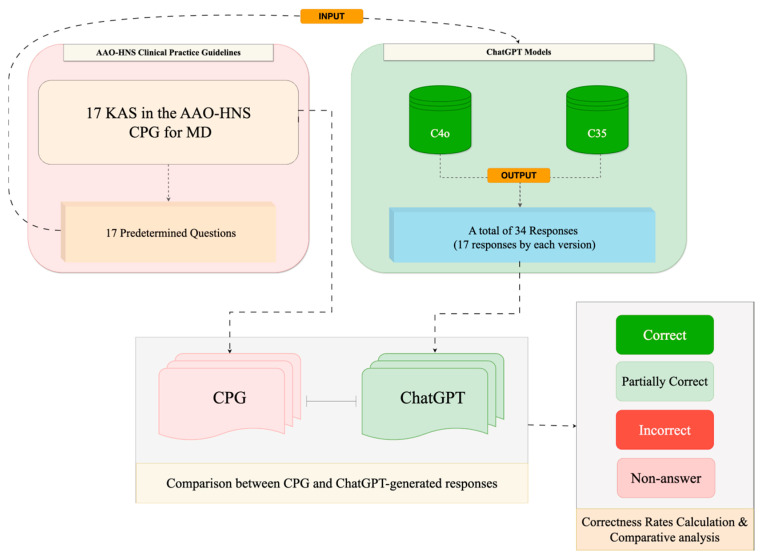
The workflow of the study. AAO-HNS: American Academy of Otolaryngology–Head and Neck Surgery; KAS: key action statement; C4o: ChatGPT 4o model in Chinese; C35: ChatGPT 3.5 model in Chinese; CPG: clinical practice guidelines.

**Figure 2 diagnostics-15-02006-f002:**
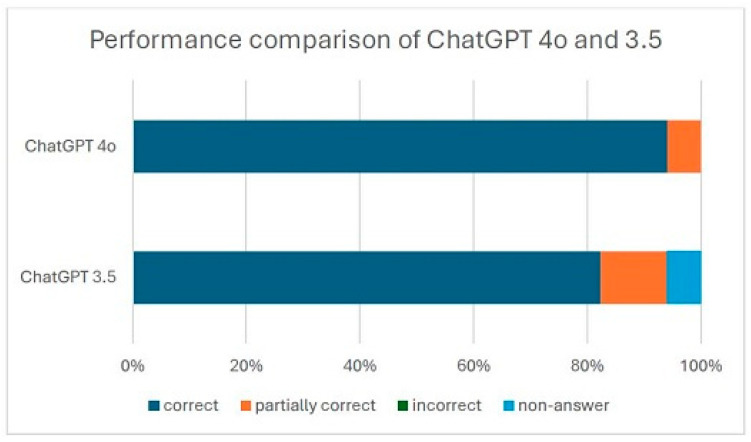
Comparison of ChatGPT 3.5’s vs. 4.0’s performance.

**Figure 3 diagnostics-15-02006-f003:**
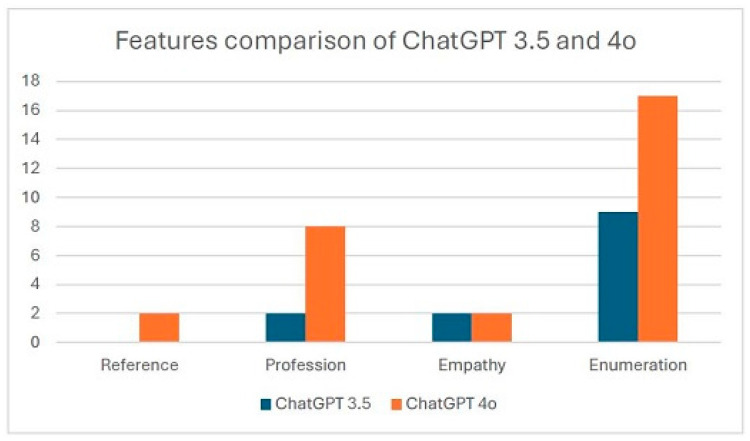
Comparison of response features between ChatGPT 3.5 and 4o.

**Table 1 diagnostics-15-02006-t001:** A comparison of the accuracy and characteristics of ChatGPT-3.5 and 4.0.

Category	ChatGPT-3.5	ChatGPT-4.0
Correct Answers	14/17 (82.4%)	16/17 (94.1%)
Partially Correct Answers	2/17 (11.8%)	1/17 (5.9%)
Non-Answers	1/17 (5.9%)	0/17 (0%)
Incorrect Answers	0	0
Overall Correctness (Correct + Partially Correct)	94.1%	100%
Statistical Comparison	Non-inferior (*p* = 0.6012, Fisher’s exact test)	Higher correctness, but not statistically significant
Citation of References	None	Frequently cited (e.g., AAO-HNS, ICHD)
Guideline Adherence	Needed follow-up queries for complete answers in 1 response	Answered accurately on first attempt
Empathy in Communication	Promoted open, personalized communication	Added clarity, reassurance, and patient empowerment
Recommendation of Specialists	2 mentions of consulting otolaryngologists/therapists	8 mentions of consulting specific medical specialists
Response Structure	Mixed and less structured	Clearly structured, enumerated, and organized
Clarity and Readability	Generally concise, but sometimes incomplete	Clear and comprehensive, but occasionally verbose

## Data Availability

The article encompasses the original contributions outlined in the study. Additional enquiries may be sent to the corresponding author.

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
