# Peer review of "Evaluating ChatGPT’s Concordance with Clinical Guidelines of Ménière’s Disease in Chinese"

_diagnostics, 2025, doi:10.3390/diagnostics15162006_

Round 1

Reviewer 1 Report

Comments and Suggestions for Authors

We are entering an era where AI models are increasingly applied in healthcare. Studies like this are potentially valuable for identifying areas of improvement and optimizing patient care and self-management. This manuscript has two major strengths: firstly, it focuses on Ménière’s disease, a chronic and debilitating condition; secondly, it is conducted in Chinese, the most widely spoken language in the world.

I have two major comments regarding the methodology:

The example question, “Should clinicians carry out audiometric tests on all patients with a suspected clinical diagnosis of MD?”, is not appropriate. This type of question is directed toward doctors, not patients. The same applies to another example in the results section: “Should clinicians educate patients with Ménière’s disease about the natural history, measures for symptom control, treatment options, and outcomes?” Since the purpose of the study is to explore the feasibility of integrating ChatGPT into patient education, these examples are not particularly relevant. I recommend replacing them with more suitable examples that are properly tailored to patient-centered education.

Ménière’s disease is a chronic and debilitating condition often associated with stress, and many patients may require psychological support. While AI can assist in providing counseling, the dichotomous (Yes/No) format of the questions used in this study limits the possibility of even a basic qualitative assessment. I therefore recommend including a few open-ended or more nuanced questions that allow for qualitative evaluation.

Author Response

Comment 1: [The example question, “Should clinicians carry out audiometric tests on all patients with a suspected clinical diagnosis of MD?”, is not appropriate. This type of question is directed toward doctors, not patients. The same applies to another example in the results section: “Should clinicians educate patients with Ménière’s disease about the natural history, measures for symptom control, treatment options, and outcomes?” Since the purpose of the study is to explore the feasibility of integrating ChatGPT into patient education, these examples are not particularly relevant. I recommend replacing them with more suitable examples that are properly tailored to patient-centered education.]

Response 1: [Thank you for pointing this out. After reviewing our manuscript, a few changes have been revised. Since our study is based on the key action statements (KAS) in the AAO-HNS clinical practice guidelines (CPG), which are established primarily to increase rates of accurate diagnosis, improve symptom control with appropriate treatments, and reduce inappropriate use of medications, procedures, or testing. In other words, KAS is drafted for healthcare professionals, not for patients. Considering this, we should adjust our primary goal from "improving patient education" to "enhancing clinical operations/workflows" for better clarity of the major focus of our study.]

Comment 2: [While AI can assist in providing counseling, the dichotomous (Yes/No) format of the questions used in this study limits the possibility of even a basic qualitative assessment. I therefore recommend including a few open-ended or more nuanced questions that allow for qualitative evaluation. Thank you for your suggestion.]

Response 2: [Agreed. Unfortunately, we have attempted this method prior to using yes/no questions, and we found several problems about this. Firstly, when asked with open-ended questions, the chatbot might give off-topic or ambiguous answers, and it is difficult to conduct further qualitative analysis since our major focus is "clinical concordance". Also, it is challenging to perform comparative analysis with open-ended questions and for statistical analysis as well. Lastly, open-ended questions are more susceptible to the subjectivity of different researchers' opinions. To minimize the problem mentioned, we selected yes/no questions instead of open-ended questions eventually. Thank you for your suggestion.]

Reviewer 2 Report

Comments and Suggestions for Authors

Dear authors,

This is a relevant manuscript and important topic. Overall, the manuscript is well-structured and methodologically sound, but several points require clarification and enhancement. I leave some suggestions:

  1. While the manuscript states that 17 questions derived from the AAO-HNS guidelines were used, neither the main text nor the supplementary materials provide the full list of these questions and corresponding ChatGPT responses. For full reproducibility and transparency, please include a detailed supplementary table containing:
    • Each question in English and Chinese (at present there are only 2 examples in Chinese).
    • The verbatim responses from both ChatGPT versions.
    • The assigned correctness classification (correct, partially correct, non-answer).
    • Reviewer comments if available.
  1. The authors state that responses were independently reviewed and then a consensus was reached. To strengthen the methodological rigor, I recommend reporting an inter-rater agreement index (e.g., Cohen’s or Fleiss’ kappa) to quantify initial consistency among reviewers prior to consensus. This would provide transparency regarding the subjective evaluation process.
  2. The manuscript highlights the importance of readability for patient education but does not provide a formal assessment. A qualitative analysis (e.g., using linguistic features such as sentence length, use of medical jargon, or enumerative structures) would strengthen this aspect. At minimum, discuss potential surrogate measures or methods used in prior Chinese-language health literacy studies.
  3. The study is confined to Ménière’s disease, which limits generalization. While the authors acknowledge this, expanding the discussion on how this framework could be adapted to other medical conditions would be valuable. Consider including suggestions for future research directions or pilot data from other disease areas (if available).
  4. In the Introduction, consider expanding on the current lack of Chinese-language patient education resources specific to otology and balance disorders.
  5. In the Discussion, better emphasize the potential ethical implications of AI-generated medical information (e.g., misinformation risk, liability, bias). What would be the risks or practical implications of incorrect answers in this case?
  6. The tendency of ChatGPT-4.0 to provide overly detailed or redundant responses could be illustrated with explicit examples, in-text or in the supplements.

Hope these suggestions help improve the manuscript’s clarity, methodological rigor, and practical relevance.

Kind regards

Author Response

Comment 1:[While the manuscript states that 17 questions derived from the AAO-HNS guidelines were used, neither the main text nor the supplementary materials provide the full list of these questions and corresponding ChatGPT responses. For full reproducibility and transparency, please include a detailed supplementary table containing:

Each question in English and Chinese (at present there are only 2 examples in Chinese).

The verbatim responses from both ChatGPT versions.

The assigned correctness classification (correct, partially correct, non-answer).

Reviewer comments if available.

]

Response 1:[Thank you for your recommendation. We will upload a supplement file to include the questions and responses.]

Comment 2:[The authors state that responses were independently reviewed and then a consensus was reached. To strengthen the methodological rigor, I recommend reporting an inter-rater agreement index (e.g., Cohen’s or Fleiss’ kappa) to quantify initial consistency among reviewers prior to consensus. This would provide transparency regarding the subjective evaluation process.]

Response 2:[Thank you for pointing this out. Inter-rater reliability was reported in the updated manuscript in the second to last sentence in the first paragraph of the results section.]

Comment 3:[The manuscript highlights the importance of readability for patient education but does not provide a formal assessment. A qualitative analysis (e.g., using linguistic features such as sentence length, use of medical jargon, or enumerative structures) would strengthen this aspect. At minimum, discuss potential surrogate measures or methods used in prior Chinese-language health literacy studies.]

Response 3:[Thank you for pointing this out. We have added this part in the fourth paragraph of the discussion part in our updated manuscript.]

Comment 4[The study is confined to Ménière’s disease, which limits generalization. While the authors acknowledge this, expanding the discussion on how this framework could be adapted to other medical conditions would be valuable. Consider including suggestions for future research directions or pilot data from other disease areas (if available).]

Response 4:[Thank you for pointing this out. However, discussion regarding similar study framework on different medical topics and suggestion for future research directions are already presented in the discussion part. In short, the comparative analysis showed GenAI's potential as a multilingual, multidisciplinary clinical assistant for both healthcare providers and patients.]

Comment 5:[In the Introduction, consider expanding on the current lack of Chinese-language patient education resources specific to otology and balance disorders.]

Response 5:[Thank your for pointing this out. However, many patient education resources introducing otology and balance disorders in Chinese are readily available on the Internet. It is the established, evidence-based clinical practice guidelines specific in Chinese that we are lacking.]

Comment 6:[In the Discussion, better emphasize the potential ethical implications of AI-generated medical information (e.g., misinformation risk, liability, bias). What would be the risks or practical implications of incorrect answers in this case?]

Response 6:[Thank you for pointing this out. We have added some words to our manuscript, from line 276 to 283, the 9th paragraph in Discussion.]

Comment 7:[The tendency of ChatGPT-4.0 to provide overly detailed or redundant responses could be illustrated with explicit examples, in-text or in the supplements.]

Response 7:[Thank you for pointing this out. A full list of the responses of ChatGPT models is accessible to the newly uploaded supplement file.]

Reviewer 3 Report

Comments and Suggestions for Authors

By every review I do try to answer the following questions:

Is the topic relevant to the field? In this case, I see no relevance. The subject is the use of AI but I do not see any scientific significance. I would like to ask the authors, why should a clinical doctor use such a methodology?

What does the article add compared to other publications? I do not see any relevance in this article. In my opinion there is no specific improvement. 

The conclusions are not consistent with the evidence and arguments presented. For such a purpose more subjects should be examined. 

The statistical analysis is poor. There should be more diagramms and tables. 

Author Response

Comment 1:[Is the topic relevant to the field? In this case, I see no relevance. The subject is the use of AI but I do not see any scientific significance. I would like to ask the authors, why should a clinical doctor use such a methodology?]

Response 1:[Thank you for pointing this out. This study is an extension of prior studies evaluating the performance of GenAI models like ChatGPT on providing accurate medical information to both patients and healthcare workers, based on established clinical practice guidelines (CPG). Due to the lack of well-recognized CPG in Chinese-speaking societies, we adapted questions from the CPG established by the AAO-HNS, which is the most-recognizable academy in the field of otolaryngology in the world. With the exponential growth of the power of AI, it is best for us to learn how to harness it, which includes knowing its strengths and weaknesses. If you are interested in studies with similar methodologies, here are some related studies:

  • Ho, R. A., Shaari, A. L., Cowan, P. T., & Yan, K. (2024). ChatGPT responses to frequently asked questions on Ménière's disease: a comparison to clinical practice guideline answers. OTO open, 8(3), e163.
  • Maksimoski, M., Noble, A. R., & Smith, D. F. (2024). Does Chat GPT Answer Otolaryngology Questions Accurately?. The Laryngoscope, 134(9), 4011-4015.
  • Moise, A., Centomo-Bozzo, A., Orishchak, O., Alnoury, M. K., & Daniel, S. J. (2023). Can ChatGPT guide parents on tympanostomy tube insertion?. Children, 10(10), 1634.
  • Lee, S. J., Na, H. G., Choi, Y. S., Song, S. Y., Kim, Y. D., & Bae, C. H. (2023). Accuracy of the Information on Sudden Sensorineural Hearing Loss From Chat Generated Pre-Trained Transformer. Korean Journal of Otorhinolaryngology-Head and Neck Surgery, 67(2), 74-78.]

Comment 2:[What does the article add compared to other publications? I do not see any relevance in this article. In my opinion there is no specific improvement.]

Response 2:[Thank you. As we have mentioned in the discussion part, “This study is the first to incorporate Chinese into a comparative analysis of ChatGPT’s performance between the 3.5 and 4.0 models on the topic of MD.” While the majority of the studies with similar methodologies were conducted in English, we decided to test whether the GenAI’s performance in Chinese-based text generation was as good as that in English.]

Comment 3:[The conclusions are not consistent with the evidence and arguments presented. For such a purpose more subjects should be examined.]

Response 3:[Thank you. We acknowledge your concern regarding the consistency between our conclusions and the evidence presented. Given our current study's exploratory nature and limited sample size, the conclusions should indeed be interpreted cautiously. In future studies, we plan to include more subjects and expand our analysis to enhance the robustness of our findings and to better align the evidence with our conclusions.]

Comment 4:[The statistical analysis is poor. There should be more diagrams and tables.]

Response 4:[Thank you. Our primary focus in this study was on clinical concordance with the CPG, which limited the extent of statistical analysis presented. However, we agree that incorporating additional diagrams, tables, and more comprehensive statistical analyses would strengthen the presentation and clarity of our findings. We will aim to include these improvements in future studies.]

Reviewer 4 Report

Comments and Suggestions for Authors

This is an interesting paper that discusses the application of tow AI programs that can assist physicians specifically with patients who present with Ménière’s disease.    The authors claim that their results with ChatGPT 3.5 and 4.o is accurate to have some value.   They base this on using the AAO guidelines with the 17 questions and then divide the answers into 4 categories to determine the accuracy, based on the consensus of 3 experts.   My concern is that there is s certain arbitrary designation that forms the categories and it is not clear how these experts reached a consensus.  Also there is no table listing the 17 questions.

It is otherwise a well written and researched subject with limitations mentioned in the of only viewing a specific disease entity as well as using only one ChatGPT as there are now other AI programs available.

Author Response

Comment 1:[This is an interesting paper that discusses the application of tow AI programs that can assist physicians specifically with patients who present with Ménière’s disease.    The authors claim that their results with ChatGPT 3.5 and 4.o is accurate to have some value.   They base this on using the AAO guidelines with the 17 questions and then divide the answers into 4 categories to determine the accuracy, based on the consensus of 3 experts.   My concern is that there is a certain arbitrary designation that forms the categories and it is not clear how these experts reached a consensus.  Also there is no table listing the 17 questions.]

Response 1:[Thank you for pointing this out. Reviewer responses were compared for congruency after both reviewers finished evaluation. Inter-rater reliability prior to reaching a consensus was calculated using Microsoft Excel and presented in the updated manuscript. After thorough discussion, the reviewers reached a consensus on the clinical accordance of the GenAI-generated-texts with established CPGs. The questions are listed in the newly uploaded supplement file.]

Comment 2:[It is otherwise a well written and researched subject with limitations mentioned in the of only viewing a specific disease entity as well as using only one ChatGPT as there are now other AI programs available.]

Response 2:[Thank you. In future studies, we are planning to evaluate the performance of different GenAI models in more diverse topics as well.]

Round 2

Reviewer 1 Report

Comments and Suggestions for Authors

I believe that the authors made a clever change of the reported purpose of their article and their work now supports their purpose. 

Regarding their response on my comment on open ended questions I believe it is important to be included in the discussion of the manuscript since as they say "the chatbot might give off-topic or ambiguous answers" If this is true then it is of interest for the clinicians 

Author Response

Comment 1: [Regarding their response on my comment on open ended questions I believe it is important to be included in the discussion of the manuscript since as they say "the chatbot might give off-topic or ambiguous answers" If this is true then it is of interest for the clinicians ]

Response 1: [Thank you for pointing this out. What you mentioned are the words that we described the characteristics of ChatGPT-3.5 in line 199, and we cannot agree with you more. Unfortunately, ChatGPT-3.5 has been deprecated by the OpenAI due to the rapid evolvement of its models, so we are unable to test ChatGPT-3.5 with open-ended questions anymore. We will take your comments into consideration in our future studies.]

Reviewer 2 Report

Comments and Suggestions for Authors

Dear authors,

Thank you for the careful revisions and for substantially improving the manuscript.

The structure is now clearer, and the comparative analysis between GPT-3.5 and 4.0 is well-articulated. However, a few important issues remain:

  • Supplementary material: In the initial review, it was requested that the full responses generated by ChatGPT be provided in both Chinese and English. The current supplementary file includes only the Chinese versions. To ensure transparency, reproducibility, and accessibility to a broader readership, the English translations should be included as well—either alongside the Chinese or in a separate file.

  • Language clarity: A few sections would benefit from final language refinement to improve fluency and remove awkward constructions (e.g., lines 24, 57, 260–266).

Author Response

Comment 1: [Supplementary material: In the initial review, it was requested that the full responses generated by ChatGPT be provided in both Chinese and English. The current supplementary file includes only the Chinese versions. To ensure transparency, reproducibility, and accessibility to a broader readership, the English translations should be included as well—either alongside the Chinese or in a separate file.]

Response 1: [Thank you for pointing this out. A supplementary file is uploaded for your request.]

Comment 2: [Language clarity: A few sections would benefit from final language refinement to improve fluency and remove awkward constructions (e.g., lines 24, 57, 260–266).]

Response 2: [Thank you. Unsure of which words or sentences you would recommend to change, some words in line 24, 57 and 263 are modified.]

Reviewer 3 Report

Comments and Suggestions for Authors

Though the text is sign ificantly improved, in my opinion further improvement is required. I present my points step by step Introduction==> The second and the third paragraphs shall be combined in onme. Methods==> Further information such as degree, kind of studies, age etc can improve thoroughly the quality of the article. Results==> more diagramms are needed. The results can be better presented. Discussion==> It is ok.k in the present form. I d not have specific comments. References==> I do not have specific suggestions.

Author Response

Comment 1: [Introduction==> The second and the third paragraphs shall be combined in one. Methods==> Further information such as degree, kind of studies, age etc can improve thoroughly the quality of the article. Results==> more diagramms are needed. The results can be better presented. Discussion==> It is ok.k in the present form. I d not have specific comments. References==> I do not have specific suggestions.]

Response 1:[Thank you. We have adjusted the content according to your opinion on the introduction part. However, regarding your opinion on methods, we have mentioned that this is a comparative analysis study in line 87 and 110. You can also know it is a comparative analysis in line 32 in the abstract. You kindly mentioned that it is better to include subjects' age in our methods, which is confusing since none of our subjects have an age due to the fact that they are GenAI-generated responses. For the results part, we added a table demonstrating the comparison of both the accuracy and the characteristics of the two models. Hopefully this would help. Some of the figures showing the prompting processes on GenAI's platform are presented in the supplement file.]

Reviewer 4 Report

Comments and Suggestions for Authors

Well written paper that presents new applications of AI modalities that should be of interest to our readers.

Author Response

Comment 1: [Well written paper that presents new applications of AI modalities that should be of interest to our readers.]

Response 1: [Thank you for your comment. ]

Round 3

Reviewer 3 Report

Comments and Suggestions for Authors

I have seen the changes made by the authors. I do not think that they are substantial. n any case I do not wnt tro repeat myself. The authors can see my previous suggestions. I do not think that the whole text has been changed drastically. I believe that, if there are no substantial changes, as I have already proposed, the text must be rejected. Any further discussion just waste of time. n